# Non-Abelian effects in dissipative photonic topological lattices

Midya Parto [1,6], Christian Leefmans[2,6], James Williams[1], Franco Nori [3,4,5] & Alireza Marandi [1,2] ✉

Topology is central to phenomena that arise in a variety of fields, ranging from quantum field theory to quantum information science to condensed matter physics. Recently, the study of topology has been extended to open systems, leading to a plethora of intriguing effects such as topological lasing, exceptional surfaces, as well as non-Hermitian bulk-boundary correspondence. Here, we show that Bloch eigenstates associated with lattices with dissipatively coupled elements exhibit geometric properties that cannot be described via scalar Berry phases, in sharp contrast to conservative Hamiltonians with non-degenerate energy levels. This unusual behavior can be attributed to the significant population exchanges among the corresponding dissipation bands of such lattices. Using a one-dimensional example, we show both theoretically and experimentally that such population exchanges can manifest themselves via matrix-valued operators in the corresponding Bloch dynamics. In two-dimensional lattices, such matrix-valued operators can form non-commuting pairs and lead to non-Abelian dynamics, as confirmed by our numerical simulations. Our results point to new ways in which the combined effect of topology and engineered dissipation can lead to non-Abelian topological phenomena.

According to quantum mechanics, the dynamics of a closed system are governed by a set of unitary operators. On most occasions, however, a real physical arrangement inevitably exchanges energy with its surrounding environment–something that has traditionally been considered an adverse effect, as it produces decoherence and energy decay[1]. Yet, recent studies have shown that dissipative interactions, which occur when the elements of a system exchange information through the surrounding environment, may be used as valuable tools for shaping the responses of open systems[2–4]. Such engineered dissipation has been successfully implemented in various settings, ranging from quantum computing[5,6] and information processing[7–10] to active optical platforms to electronic and mechanical arrangements[11].

Recent years have witnessed a flurry of interest in the emerging field of topological physics[12,13]. One of the most prominent examples of topological behavior is a set of materials exhibiting nonzero topological invariants which are endowed with inherent robustness against local disorders[14]. This type of topological protection also occurs in physical settings beyond condensed matter physics and has led to unidirectional transport and robust features in optics[15–19], cold atoms[20,21], mechanics[22], and acoustics[23,24]. While topological phenomena were originally studied in closed systems, recent works on topology in *open* systems have led to a host of intriguing effects[25–30]. For instance, the interplay between topology and dissipation/gain has been utilized to develop robust and efficient coherent light sources[26,31–36]. Other studies include the emergence of topological phases from purely dissipative interactions in the absence of Hamiltonian couplings[37–39], as well as extending the bulk-edge correspondence and topological band theory to open and non-Hermitian settings[40–45].

[1]Department of Electrical Engineering, California Institute of Technology, Pasadena, CA, USA. [2]Department of Applied Physics, California Institute of Technology, Pasadena, CA, USA. [3]Theoretical Quantum Physics Laboratory, RIKEN, Wakoshi, Saitama, Japan. [4]RIKEN Center for Quantum Computing, Wakoshi, Saitama, Japan. [5]Department of Physics, University of Michigan, Ann Arbor, MI, USA. [6]These authors contributed equally: Midya Parto, Christian Leefmans. ✉e-mail: marandi@caltech.edu

Gauge fields are pivotal to the understanding of topological phenomena that arise, for instance, in the Pancharatnam-Berry phases first introduced in polarization optics. On many occasions, these gauge fields belong to the Abelian class which give rise to closed-loop evolution operators in the parameter space that commute with each other. This simple picture changes drastically in more complex scenarios which involve non-Abelian gauge fields where the corresponding operators along different paths are no longer commutative and can be utilized to obtain universal gates for topological quantum computing[46]. Although Abelian gauge fields have been widely used for characterizing topological states[47–51], their non-Abelian counterparts have largely remained unexplored. This is mainly due to the strict requirements, such as the existence of degenerate states in the underlying Hilbert space that are necessary in a conservative system to host non-commutative evolution operators[52,53]. Quite recently, non-Abelian effects and topological charges have been observed in a variety of photonic systems[54–57] involving coupled waveguide arrays[58,59] and nonreciprocal elements[60]. Despite intense research efforts in this area, studies have exclusively focused on systems with conservative couplings. In contrast, we show how the synergy between *topology* and *dissipation* can give rise to non-Abelian dynamics in the reciprocal space.

Here, we show that Lindbladians involving *dissipative couplings* can be governed by matrix-valued modified Wilson lines[52], leading to non-Abelian effects[56,59–61]. To do so, we experimentally measure nontrivial geometric phases and demonstrate signatures of non-Abelian effects in a dissipatively-coupled network of time-multiplexed photonic resonators. In contrast to conservative systems possessing nondegenerate energy levels wherein the geometric properties of the Bloch eigenstates are typically predicted by scalar Berry phases, here, significant population exchange can occur among the ensuing dissipation bands. Our simulations involving a two-dimensional honeycomb lattice illustrate how such dynamics lead to non-Abelian operators acting on the reciprocal space.

## Results

The general model of a Markovian open lattice is described by the Lindblad master equation:

$$\frac{d}{dt}\hat{\rho} = \mathcal{L}\hat{\rho} \equiv -i[\hat{H},\hat{\rho}] + \sum_j \mathcal{D}[\hat{L}_j]\hat{\rho}, \qquad (1)$$

where $\hat{\rho}$ represents the system density operator, $\hat{H}$ signifies the system Hamiltonian and $\mathcal{D}[\hat{L}_j] = \hat{L}_j\hat{\rho}\hat{L}_j^\dagger - 1/2\{\hat{L}_j^\dagger\hat{L}_j,\hat{\rho}\}$ is the dissipator resulting from the nonlocal jump operators $\hat{L}_j$ acting upon the lattice site $j$. When $\hat{H} = 0$, the lattice sites only exchange energy via the dissipators $\mathcal{D}[\hat{L}_j]$. Such purely dissipative couplings can be engineered to map the energy spectra of arbitrary tight-binding Hamiltonians into the decay rates of the corresponding open

system[38,62]. In particular, by properly choosing $\mathcal{D}[\hat{L}_j]$, the Lindbladian of Eq. (1) supports Bloch eigenstates characterized by bands of dissipation rates in the reciprocal space (see Supplementary Part 1).

For our experiments, we use a time-multiplexed photonic resonator network depicted schematically in Fig. 1. This time-multiplexed network consists of a main fiber loop (the "Main Cavity"), which supports $N = 64$ resonant pulses separated by a repetition period, $T_R$. Each pulse represents an individual resonator associated with the annihilation (creation) operators $\hat{c}_j^{(\dagger)}$ in Eq. (1). In addition, in order to realize the jump operators $\hat{L}_j$, we construct delay lines to dissipatively couple the individual pulses. Each delay line is equipped with intensity modulators that control the strengths of these couplings (see Fig. 1). It should be emphasized that the setup used here has two important distinctions compared to the one used in[38] that enables us to experimentally investigate geometric properties and non-Abelian signatures associated with topological dissipative bands. First, the present work uses a homodyne detection scheme to resolve the phase information encoded in the optical fields, as necessary to measure geometric phases. Second, to probe the geometric properties of the dissipation bands, we apply a constant force in the reciprocal space which translates into a closed-loop evolution defined on the Brillouin zone in the Bloch-momentum space. This type of evolution results in periodic revivals in the dynamics of the optical fields across the lattice which are known as Bloch oscillations (BO). To achieve this, we implement the Hamiltonian $\hat{H}_{BO} = \boldsymbol{F} \cdot \hat{\boldsymbol{r}}$, where $\boldsymbol{F}$ represents a constant effective force along the reciprocal lattice direction $\boldsymbol{r}$. In Supplementary Part 2 we show that this Hamiltonian can be approximated by a pulse-to-pulse linear phase gradient implemented by a phase modulator in the main cavity of our network.

To experimentally demonstrate BOs, we first construct a 1D lattice (see Supplementary Part 3) with uniform, nearest-neighbor dissipative couplings (Fig. 2a). The jump operators in the master equation describing this lattice are given by $\hat{L}_j = \sqrt{\Gamma}(\hat{c}_j + \hat{c}_{j+1})$ (see Supplementary Part 1). We excite a single lattice site in the network and observe its evolution under different pulse-to-pulse phase gradients, which correspond to different BO Hamiltonians $H_{BO}$. We first investigate the dynamics in the absence of a phase gradient (i.e., $\hat{H}_{BO} = 0$). In this case, the excitation undergoes dissipative discrete diffraction (Fig. 2b). We emphasize that the shape of the diffraction pattern in this figure is qualitatively different from its conservative counterparts[63] due to the dissipative couplings involved (see Supplementary Part 7). Next, we turn on the linear gradient potential associated with $\hat{H}_{BO}$. Figure 2c, d shows experimental pulse propagation measurements associated with $\phi_0 = 2\pi/8$ and $\phi_0 = 2\pi/4$, which correspond to Bloch periods of 8 and 4 network roundtrips, respectively. As evident from these figures, the presence of pulse-to-pulse phase gradients causes the excitation to undergo periodic diffraction and refocusing, which is the hallmark of Bloch oscillations.

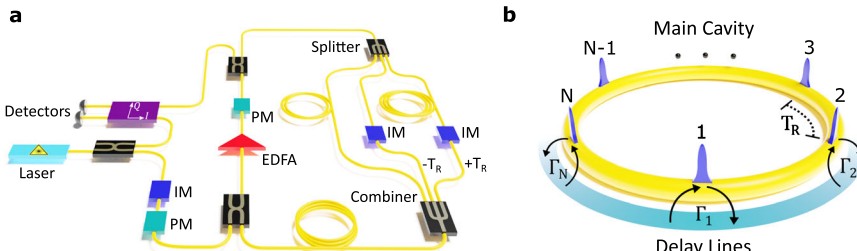

**Fig. 1 | Network of time-multiplexed resonators. a** Schematic diagram of the experimental setup used to implement dissipatively coupled resonators. An intensity modulator (IM) and a phase modulator (PM) are used in the input of the optical fiber to generate arbitrary wavefunctions defined by injected femtosecond pulses from a mode-locked laser with a repetition rate of $T_R$. An Erbium-doped fiber amplifier (EDFA) is used in the main cavity to compensate for the losses and

increase the number of measurement roundtrips. Two delay lines with smaller and larger lengths than the main cavity (corresponding to delays of $-T_R$ and $+T_R$, respectively) are used to dissipatively couple the pulses. **b** Schematic of a resonant cavity loop (yellow) which hosts $N$ pulses, each representing a resonator element in a dissipatively-coupled lattice. The delay lines (shown in green) provide the dissipative couplings with different rates between nearest-neighbor sites.

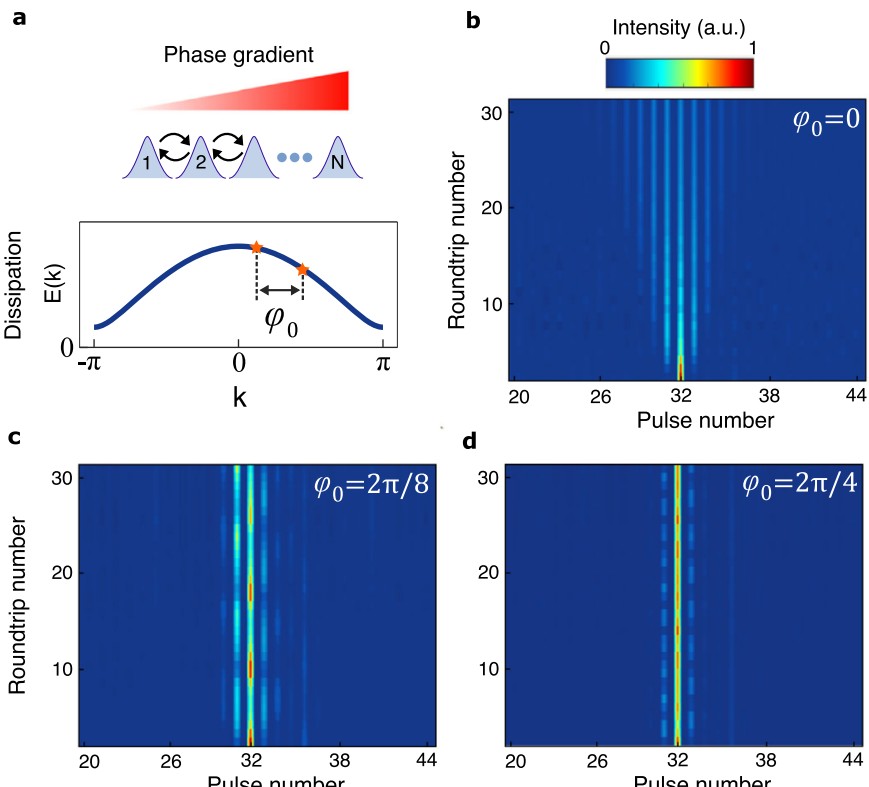

**Fig. 2 | Experimental demonstration of Bloch oscillations in a uniform, dissipatively-coupled open lattice. a** Applying a phase gradient among the pulses in the time-multiplexed network transports the associated Bloch eigenstates in the reciprocal space by a value of $\delta k = \phi_0$ per cavity roundtrip, where $\phi_0$ denotes the pulse-to-pulse phase differences induced by the intracavity phase modulator. **b** to **d,** Pulse intensity measurements associated with $\phi_0 = 0, 2\pi/8$ and $2\pi/4$, respectively. In all cases, optical power is initially launched into one lattice element (pulse

number 32). As shown in **b**, in the absence of the effective force ($\phi_0 = 0$) light undergoes dissipative discrete diffraction in the lattice. In contrast, when a nonzero phase gradient is established among the pulses, optical power exhibits an oscillatory pattern with a Bloch period equal to $N_B = 8, 4$ in **c**, **d**, respectively. In all cases, the optical power across the lattice sites is normalized in every round trip to provide a more distinct visualization of the field intensities.

Typically, in a Hamiltonian lattice with multiple energy bands, the associated Bloch states tend to remain in a single band when subject to a sufficiently weak external force $F$. Under such adiabatic conditions, the state undergoes Bloch oscillations and merely acquires a phase factor comprised of a dynamical part in addition to the geometric Berry phase associated with its energy band[64,65]. This is in sharp contrast to Lindbladian lattices exhibiting dissipation bands which in general may not be considered in isolation. In this sense, the two-band open systems studied here exhibit reciprocal-space dynamics and band topologies similar to those of Hamiltonian systems possessing quasi-degenerate energy levels[53,66]. Here, the reciprocal-space dynamics produced by the Lindbladian in Eq. (1), with $\hat{H} = \hat{H}_{BO}$ are governed by the modified Wilson line operator

$$\hat{W}'_{\boldsymbol{k}(0) \to \boldsymbol{k}(t)} = \mathcal{T} \exp \int dt \left[ \hat{\mathbf{E}}(\boldsymbol{k}(t)) + i\hat{\mathbf{A}}(\boldsymbol{k}(t)) \cdot \boldsymbol{F} \right], \quad (2)$$

where $\mathcal{T}$ indicates time ordering, $\hat{\mathbf{E}}$ is a diagonal matrix containing dissipation rates for different bands while $\hat{\mathbf{A}}$ represents the Wilczek-Zee connection matrix $\mathbf{A}_{p,q} = i\langle \phi_p(\boldsymbol{k})|\boldsymbol{\nabla}_{\boldsymbol{k}}|\psi_q(\boldsymbol{k})\rangle$ at $\boldsymbol{k}(t) = \boldsymbol{k}(0) + \boldsymbol{F}t$ corresponding to the non-Hermitian Bloch Hamiltonian associated with the system Lindbladian[52] (see Supplementary Parts 5 and 6). Here, $p, q$ represent two arbitrary bands of the system.

In order to show how the modified Wilson line in Eq. (2) can be used to experimentally characterize gauge fields and topological invariants within dissipative bands, we first consider a one-dimensional Su-Schrieffer-Heeger (SSH) lattice (Fig. 3a). To implement this, the intensity modulators within the network are programmed to realize the

staggered couplings of the SSH model. These couplings correspond to the jump operators $\hat{L}_{A,j} = \sqrt{\Gamma_A}(\hat{c}_{A,j} + \hat{c}_{B,j})$ and $\hat{L}_{B,j} = \sqrt{\Gamma_B}(\hat{c}_{A,j+1} + \hat{c}_{B,j})$ (Supplementary Part 1), where $A$ and $B$ represent the two sublattices in the structure. With these jump operators, the resulting Lindbladian exhibits a dissipative band structure that can host a topologically nontrivial bandgap (Fig. 3a). We first examine the upper-band geometric Berry phase $\theta_+$ resulting from the $\mathbf{A}_{1,1}$ component of the Wilczek-Zee connection. To do so, we initially excite the upper-band Bloch eigenstate at $k = 0$. Meanwhile, the phase modulator in the cavity is programmed to impart a pulse-to-pulse phase gradient of $\phi_0 = 2\pi/8$ to initiate Bloch oscillations in the dissipatively coupled SSH lattice. After a complete Bloch period, we measure the output of the network using homodyne detection (see Fig. 1a). As expected from Eq. (2), at this point, the observed state is in a superposition of the upper- and lower-band Bloch eigenstates (see Supplementary Part 6). Consequently, to measure the Zak phase associated with the upper band, we project the observed state into the upper-band eigenstate. Because the dissipative dynamics of our system does not impart a dynamical phase, the relative phase difference between this state and that of the originally launched pulses provides a direct measurement of the upper-band Zak phase. Figure 3b–d presents our experimentally measured values of this geometric phase in different coupling regimes of the SSH model. For $\Gamma_A = \Gamma_B$, we measure a Zak phase value of $\theta_+ = \phi_{Z0} \approx -0.02\pi$, as expected from theory (Fig. 3b). On the other hand, when $\Gamma_A \neq \Gamma_B$, our measurements show $\theta_+ = \phi_{Z1} \approx 0.47\pi$ and $\theta_+ = \phi_{Z2} \approx -0.51\pi$ for the dimerizations $D_1$ and $D_2$ depicted in Fig. 3c, d, respectively. Based on these results, the absolute value of the Zak phase in this open topological system is measured to be $\phi_Z = \phi_{Z1} - \phi_{Z2} \approx 0.98\pi$, in agreement with the

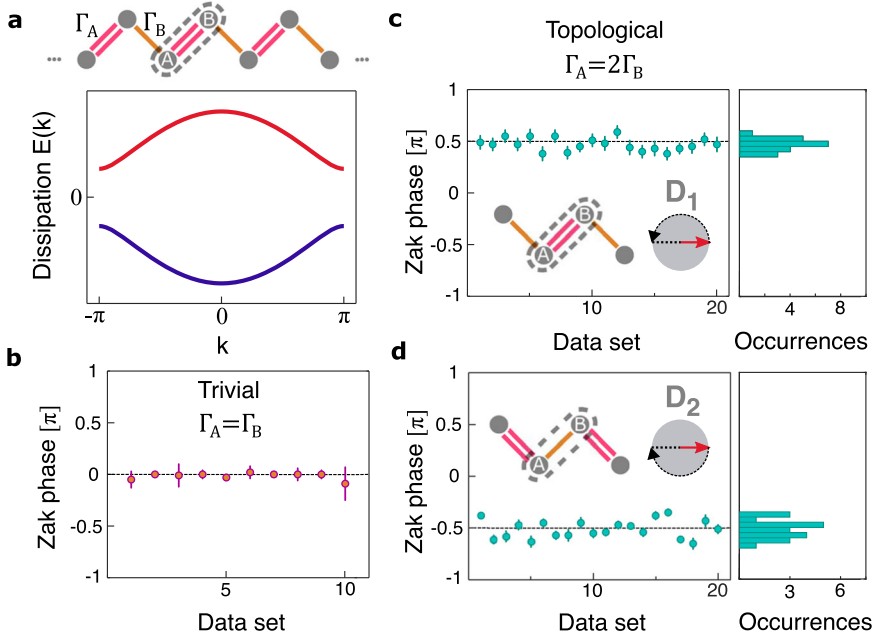

**Fig. 3 | Measuring the geometric Zak phase in a dissipative SSH model using dissipative Bloch oscillations. a** Schematic diagram of an SSH lattice with two different couplings $\Gamma_A = 2\Gamma_B$ together with its associated dissipation bands. Since the interactions among the constituent elements are arising from the corresponding dissipators, these bands represent relative gain/decay rates, with the upper-band Bloch eigenstates experiencing relative gain while the ones associated with the lower band decay faster. **b**–**d** Experimentally measured Zak phases under various coupling conditions. A trivial coupling between the lattice sites $\Gamma_A = \Gamma_B$ leads to a zero Zak phase **b**. On the other hand, when the intercell and intracell

dissipators differ, our measurements show $\phi_{Z1} \approx 0.47\pi$ and $\phi_{Z2} \approx -0.51\pi$ for the two possible dimerizations $D_1$ and $D_2$ shown in **c**, **d**, respectively. These nontrivial phases are geometrically equivalent to the counter-clockwise and clockwise windings of the upper-band Bloch eigenstates on the associated Bloch sphere, as depicted in the insets **c**, **d**, respectively. Each data set represents various unit cells (shown in dashed lines) within a single measurement, except for the two first and last units to avoid edge effects. In all cases the error bars indicate standard deviations.

theoretically expected value of $\phi_Z = \pi$ for a topologically nontrivial SSH lattice.

As discussed earlier, Eq. (2) describes different dynamics than that occurring in conservative Hamiltonian systems, since the gauge fields involved in Eq. (2) are no longer characterized by the scalar-valued Berry phases. To show this contrast, we consider a scenario wherein the Bloch eigenstate associated with the lower band of our dissipative SSH model $|\psi_-(0)\rangle$ is initially excited. Under such conditions, the off-diagonal Wilczek-Zee connections ($A_{pq}$) result in a nonzero population of the upper band in addition to the lower one. The effective force applied via $\hat{H}_{BO}$ will then transport both of these eigenstates along their corresponding bands in the reciprocal space (Fig. 4a, b). As $k$ varies between 0 and $2\pi$, each of the Bloch eigenstates $|\psi_\pm\rangle$ are multiplied by a Zak phase of $\phi_{Z1} = \pi/2$ for the $D_1$ configuration shown in Fig. 3c. Meanwhile, due to the dissipative nature of the bands, the upper-band eigenstate is relatively amplified while the lower one is attenuated more. Hence, at the end of the Brillouin zone when $k = 2\pi$, the contribution from the upper-band eigenstate that interferes with that associated with the lower one dominates the total population in this level due to its much higher amplitude. The combined effect of these inter-band transitions and parallel transports along the two bands of the dissipative SSH system is thus expected to impart a total phase of $\theta_- = -\pi/2$ to the original Eigenstate $|\psi_-(0)\rangle$ launched in the input. We note that this behavior is in stark contrast to that expected from the SSH model implemented conservatively, where both the upper and lower bands display equal geometric phase values determined by their associated Zak phases (Fig. 4a). Figure 4c presents the experimentally measured lower-band geometric phases with a mean value of $\theta_- = -0.49\pi$, confirming our theoretical predictions based on Eq. (2) (see Supplementary Part 6).

So far, we showed that the geometric properties of dissipative bands in one-dimensional lattices involve the full matrix-valued

Wilczek-Zee connections associated with the Bloch eigenstates in such systems. In this respect, our experimentally measured geometric phases provide signatures of non-Abelian effects in dissipative topological lattices. However, to define intersecting closed loops in the reciprocal space that are necessary for non-Abelian operators, one needs to consider lattices extending over more than one dimension. For this purpose, we choose a dissipative honeycomb lattice as shown in Fig. 5a. Such a lattice exhibits two distinct bands of dissipation, as shown in Fig. 5b. Starting from an arbitrary Bloch momentum in the reciprocal space $k = k(0)$, we consider two different closed loops depicted as $C_1$ and $C_2$ in Fig. 5c. By applying a force $F$ parallel to $C_i$ ($i = 1, 2$) an initial state $|\psi(k_0)\rangle$ is transformed in the reciprocal space to a new state $\hat{W}'(C_i)|\psi(k_0)\rangle$. Given that these evolutions involve time-ordered integrals with matrices that in general do not commute, we expect the final state of the system to depend on the order of such operators. Figure 5d presents our simulations using Eq. (2) for two different scenarios, where the normalized initial and final states are displayed on a Bloch sphere. These results demonstrate the non-Abelian nature of the dynamics that arise in the photonic dissipative lattices considered here. For more detailed simulations, please see Supplementary Part 10.

## Discussion

In conclusion, we have shown that non-Abelian effects can arise in a dissipatively coupled network of time-multiplexed photonic resonators. In contrast to conservative Hamiltonian systems with non-degenerate energy levels where the geometric properties are typically predicted by scalar Berry phases, here, the emerging gauge fields are in general governed by matrix-valued modified Wilson lines which may not commute for different Bloch momenta. Our measurements on the geometric Zak phases in a one-dimensional SSH model corroborate our theoretical predictions. In two dimensions, the non-Abelian nature

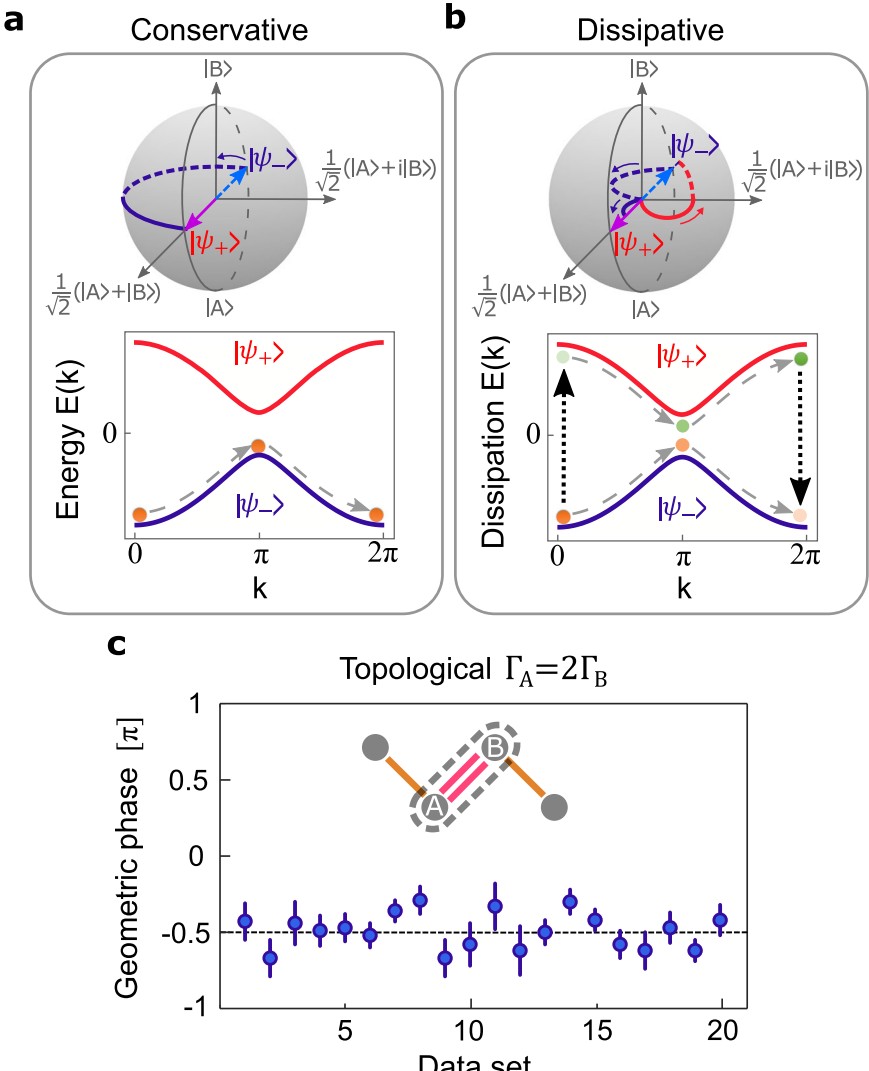

**Fig. 4 | Modified geometric phases in the presence of non-Abelian effects.**
**a** Geometric winding of the lower-band Bloch eigenstates associated with a con-
servative SSH Hamiltonian illustrated on the Bloch sphere. In this representation,
the upper and lower Bloch eigenstates are located on the equatorial plane, shown in
red and blue colors, respectively. Here, $|A\rangle$ and $|B\rangle$ represent the uniformly dis-
tributed states residing on the $A$ and $B$ sublattices, corresponding to the points
located on the south and the north poles of the Bloch sphere, respectively. The
magenta and light blue arrows represent the upper- and lower-band Bloch eigen-
states associated with the Bloch momentum $k = 0$, respectively. The lower panel
indicates reciprocal-space dynamics associated with the lower band, which in the
adiabatic regime is independent of the upper band. **b** Similar results for a dis-
sipative SSH Lindbladian are obtained from the corresponding modified Wilson line
operator (Eq. (2)). Unlike the conservative case **a**, the lower-band of a dissipative
SSH lattice is expected to exhibit a different geometric phase than that of the upper

one. This is because the dissipation bands emerging in the latter are coupled via the
off-diagonal Wilczek-Zee connections ($A_{pq}$), as illustrated in the lower panel **b**.
Hence, during a Bloch period, the upper-band eigenstates (represented by green
dots) are relatively amplified while those associated with the lower one (shown as
orange dots) experience a higher attenuation. Eventually, the state of the system at
the end of this cycle is determined by the interference between the eigenstates
associated with these two bands, which is dominated by the upper-band con-
tribution. This results in a $\pi$ phase shift in the lower-band geometric phase. The left
and right black dotted arrows represent the transfer of Bloch eigenstate popula-
tions from the lower band to the upper one and vice versa, respectively.
**c** Experimentally measured values (to be compared with Fig. 3c) indeed corrobo-
rate these theoretical predictions. In all cases the error bars indicate standard
deviations.

of the underlying dynamics can be manifested in non-commutative
operators acting on the Bloch eigenstates. Our findings unveil new
ways in which topology and engineered dissipation can interact and
lead to non-Abelian topological phenomena.

## Methods
### Experimental procedure
As discussed above, our time-multiplexed photonic resonator network
consists of a main fiber loop ("Main Cavity"), which supports 64 reso-
nant pulses separated by $T_R \approx 4$ ns, as well as two optical delay lines,
which introduce nearest neighbor dissipative couplings between the
pulses in the network. A detailed schematic of this network is shown in

Supplementary Fig. 1. To realize dissipative BOs and to measure the Zak
phases of the SSH bands, we insert intensity modulators (IMs) into the
delay lines, a phase modulator (PM) and an IM into the main cavity. The
IMs in the delay lines control the pulse-to-pulse coupling strengths,
while the intra-cavity PM in the main cavity produces a linear phase
ramp that induces Bloch oscillations. During the experiment, we use
intra-cavity IM to "Q-switch" the cavity, so that the pulses in the network
see less loss as they undergo Bloch oscillations. This allows us to operate
the system close to threshold for a brief time during the experiment and
helps to extend the number of roundtrips that we are able to observe.
Before and after the experiment, we operate the network well below the
threshold, where it is easier to actively stabilize the system.

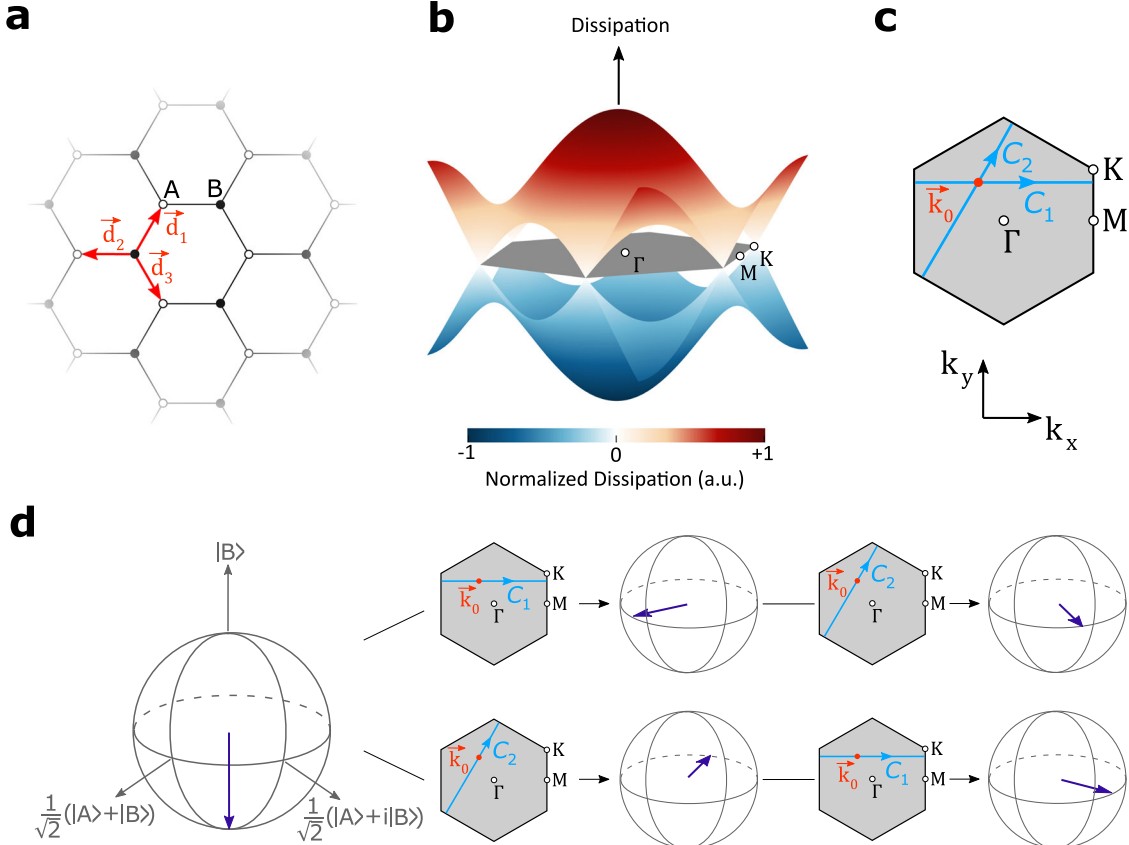

**Fig. 5 | Non-Abelian dynamics involving Bloch eigenstates in a dissipative honeycomb lattice. a** Schematic of a dissipative honeycomb lattice with two sublattices $A$ and $B$. **b** Dissipation bands associated with the Bloch eigenstates of the lattice. **c** The Brillouin zone in the reciprocal space where an initial point at $\boldsymbol{k} = \boldsymbol{k}_0$ is shown together with two different closed loops $C_1$ and $C_2$ along which the initial state is transported. **d** Simulation results displaying the initial and final states on the Bloch sphere clearly show the non-commutative nature of the modified Wilson lines defined in Eq. (2), $\hat{W}'(C_1)\hat{W}'(C_2) \neq \hat{W}'(C_2)\hat{W}'(C_1)$.

To measure dissipative BOs, we adjust the throughput of the delay lines to equalize the coupling strengths of the nearest neighbor couplings. We calibrate the phase modulator to implement a phase ramp with the desired pulse-to-pulse phase difference. Then, using IMs at the input of the main cavity, we excite a single time-slot (lattice site) in our time-multiplexed network with pulses from a mode-locked laser. With the delay line couplings and the linear phase ramp turned off, we excite this site over several roundtrips of the cavity to build up power in the network. We then stop the excitation and turn on the phase ramp and the delay line couplings. We record a time trace of the network over ~32 roundtrips with a fast photodetector and average over 10 independent time traces to generate the colormaps displayed in Fig. 2. While the signatures of Bloch oscillations are clear in this figure, we observe that asymmetry in the delay line couplings and imperfections in the linear phase ramp degrade the fidelity of the dissipative Bloch oscillations over many roundtrips.

For our geometric phase measurements, we reconstruct the roundtrip-to-roundtrip phase evolution of the network by detecting the pulses in the system with an optical hybrid coupler and two balanced detectors (see Supplementary Part 3). To detect roundtrip-to-roundtrip phase drift during our experiments, we populate 32 of the 64 time slots within the cavity with reference pulses. Coupling between these reference pulses is suppressed with the delay line IMs, and we use the mean phase of the reference pulses as a reference on each roundtrip of the experiments. These 32 reference pulses are also decoupled from the 32 remaining experiment time slots, which are used to measure the geometric phases.

To measure the Zak phase associated with the dissipative SSH model, we apply a linear phase ramp to the experiment time slots with

the intra-cavity PM, and we program the delay line IMs to couple the experiment time slots with the staggered couplings of the SSH model. Using the IMs at the input to the main cavity, we populate the reference time slots with a uniform stream of pulses and inject a Bloch state from either the upper or lower bands into the experiment time slots. We monitor the evolution of the pulses over a full Bloch period so that we can compare the initial excitation with the state of the network after one Bloch cycle.

### Data analysis
**Bloch oscillations.** For the three phase gradients used in our BO experiments, $\Delta\phi = 0$, $\Delta\phi = 2\pi/4$, $\Delta\phi = 2\pi/8$, we average the response of the network over 10 recorded traces. To better visualize the roundtrip-to-roundtrip dynamics of the system, we normalize the power in the network during each roundtrip such that the sum of the peak powers of the pulses is unity. Normalizing the power in this manner elucidates the oscillatory dynamics produced by the presence of a phase gradient because it clearly reveals the spreading and relocalization of the power.

**Geometric phase measurements.** To determine the geometric phase acquired in each band, we record the in-phase and quadrature components of the pulses as the system undergoes dissipative BOs. Because the 32-site chain used for our measurements has open boundary conditions, we neglect the dynamics experienced by the pulses at the edges. After a full Bloch period, we project the observed state into the upper and lower bands of the SSH model. For an initial excitation in the upper (lower) band, the phase acquired by the projection of this state into the upper (lower) band indicates the

geometric phase acquired within the corresponding band. We perform this procedure for an ensemble of independent measurements, and in Figs. 3 and 4, we plot the means and sample standard deviations of each measurement. From these measurements, we construct estimators for the means of the measured geometric phases, and we state these values in the main text.

## Data availability
The data used to generate the plots and results in this paper is available from the corresponding author upon reasonable request.

## Code availability
The code used to analyze the data and generate the plots for this paper is available from the corresponding author upon reasonable request.

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

## Acknowledgements

We thank Anna Soper for her assistance in the experiments. We acknowledge support from ARO Grant No. W911NF-18-1-0285 and NSF Grants Nos. 1846273 and 1918549. We wish to thank NTT Research for their financial and technical support. F.N. is supported in part by the Japan Science and Technology Agency (JST) via the Quantum Leap Flagship Program (Q-LEAP), the Japan Society for the Promotion of Science (JSPS) via the Grants-in-Aid for Scientific Research (KAKENHI) Grant No. JP20H00134, the Asian Office of Aerospace Research and Development (AOARD) via Grant No. FA2386-20-1-4069, and the Foundational Questions Institute Fund (FQXi) via Grant No. FQXi-IAF19-06.

## Author contributions

M.P., C.L., and A.M. conceived the idea and devised the experiments and the underlying theory. M.P., C.L., and J.W. constructed and performed the experiments. F.N. provided additional insights. All authors contributed to analyzing the data and writing the manuscript. A.M. supervised the project.

## Competing interests

The authors declare no competing interests.
