## [Peer review file · Nature Communications]

REVIEWER COMMENTS

Reviewer #1 (Remarks to the Author):

In this paper, the authors demonstrate non-Abelian effects in non-Hermitian systems with dissipative couplings. They have implemented the dissipatively coupled resonators using time-multiplexed photonic optical cavities. Then they compared the experimental results with theoretical calculations based on the Lindblad equation and matrix-valued modified Wilson lines.

I find the paper very interesting, and I think it contains new physics and its clear demonstration. It is also very timely as non-Abelian properties in topological systems have recently drawn significant attention in the related fields. In addition, the proposal in the paper is original as it proposes dissipative coupling as a new way to demonstrate non-abelian nature, which has not been reported before as far as I am aware. The conclusions are well-supported by the experimental and theoretical results. Therefore, I recommend that the paper be accepted for publication in Nature Communications after addressing the following comments.

1. Regarding the 2D honeycomb system Fig. 5, can the authors comment on how to implement it experimentally? Whereas the 1D SSH system can be more easily understood as the pulses are separated by the same interval in the time domain, extending the idea to 2D lattice systems may not be straightforward.
2. As shown in Fig. 5d, the non-Abelian nature can be realized with the 2D honeycomb lattice. However, it is unclear whether the non-commuting operation can also be realized in the dissipative 1D SSH lattice. Can the authors comment on whether the having a 2D system is necessary to realize well-defined non-commuting operators and whether it was chosen for a more effective presentation?
3. In the bottom panel of Fig. 4b, please explain what two vertical black arrows mean. As written in the caption, they seem to be related to the off-diagonal Wilczek-Zee connections, but it is not clear how the direction is determined and the physical meaning of the direction. Can the direction be associated with the signs of the off-diagonal terms?
4. A minor comment: please define the abbreviation of BO in the main text when it appears first for readers who are not familiar.

Reviewer #2 (Remarks to the Author):

In this manuscript, the authors study the presence of non-abelian effects in their previously presented resonator networks with dissipative couplings. Non-abelian effects are rare effects that could have an impact in topological quantum computing. This study is not the first study of non-abelian effects in topological photonics platforms, but it is the first in a purely dissipative topological photonic platform. The measurements are convincing, and the figures are excellent.

However, I have two concerns:

1. The overlap with the authors' earlier work. Both the experimental set up and the results show a very strong overlap with earlier works from the authors. For instance, the results presented up to Fig. 3 in the current measurement seem to have already been presented in Ref. 38, except for the direct measurement of the Zak phase (however the measurement of the Zak phase just confirms that the results in Ref. 38 were indeed topological, a claim already made in Ref. 38).

It looks like the novelty lies in Fig. 4, where the authors show the modification of the geometric phase of the lower band associated with the excitation of the eigenstate of lower band of the dissipative SSH. If I understand correctly, these effects were always there, including in Ref. [38], however the new measurements of the geometric phase reveal the underlying non-abelian effects. Revealing the presence of non-abelian effects is a valid and important result, but at the moment it is just hard to understand whether this is just a deeper look into previously observed effects or whether the physical effects observed here are different. This needs to be clarified, as well as the explicit differences in the experimental setup and their importance.

2. Important concepts are missing from the paper and the writing is obscure. Most notably, the authors do not explain what non-abelian effects are or why are they important, despite the fact that this is the main point of their manuscript. On a related note, the abstract reads, "Here, we show that resonator networks with dissipative couplings can be governed by matrix-valued modified Wilson lines, leading to non-Abelian effects", which is excessive jargon for the abstract. If this was a more specialized journal, maybe this would be OK, but for Nature Communications, I think this has to be fixed.

The authors launch into demonstrating Bloch oscillations without giving the reader any context on why they are doing that (also, the acronym BO is used without having been defined anywhere) .

The statements "... the shape of the diffraction pattern in this figure is qualitatively different from its conservative counter parts [61] because it involves constructive interference among the lattice elements due to the dissipative couplings involved" is unclear. Constructive interference is also a crucial aspect of discrete diffraction in conservative systems, so this statement on its own does not clarify anything.

So, in summary, the paper would need to be written in a more clear language, avoiding ambiguities and being upfront about the novelty of the paper, as well as avoiding excessive jargon and giving the reader enough context.

Reviewer #3 (Remarks to the Author):

The authors construct a one-dimensional pulse array with a splitter, a combiner, and properly chosen delay lines in this work. The delay lines introduce the jump operator, and the Lindblad master equation describes the dynamics of such a system. The authors first demonstrate the Bloch oscillation with a phase gradient introduced by a phase modulator (Fig. 2) and then measure the Zak phase inside a dissipative SSH model (Fig. 3). Here, the higher band is amplified while the lower band is attenuated. Due to the presence of the Wilczek-Zee connection matrix, there is some possibility that a mode can jump to the higher band even if it is initially excited at the lower band, and vice versa. Thus, if one measures the Zak phase of the lower band, one will unavoidably get the information of the higher band (Fig. 4). This is, technically speaking, not a non-abelian effect. Figure 5 tries to provide some information about how a non-abelian effect is possible. The title, abstract, and introduction try to argue that this system is non-abelian, but as said before, with no clear evidence to demonstrate the non-abelian behavior. Figure 5 finally returns to the focus (non-abelian) but with no experimental evidence, not even numerical simulations. Thus, I will not support its publication.

I think the following comments might help improve this work.

1. Figure 1. Need more details on why the combination of a splitter, a combiner, and the other two intensity modulators (IMs) corresponds to the jump operator. Supplementary Material Sec. 1 directly gives the results but has not explained why. For example, what is the relation between the modulation profile at the IMs and Γ ? How is the IM programmed and synchronized to realize the dimerized coupling of the SSH model?
2. The third paragraph on page 5. The abbreviation BO appears without definition. A few more words about what Bloch Oscillations are might be better here. The full name Bloch Oscillation appears at the end of the first paragraph on page 7.
3. The second paragraph on page 7. "In this sense, the multiband open ...". Why multiband? Or just only two bands?
4. Figure 3(b) shows that $\Gamma_A = \Gamma_B$ corresponds to a trivial case where there is no gap between the two bands in Fig. 3a. Figures 3(c) and (d) correspond to topological cases where there is a gap (for imaginary energy). Why here is $\pm \pi/2$, not π and 0 for the typical Hermitian SSH model. Meanwhile, in a standard Hermitian SSH model, whether a Hamiltonian is topological or trivial depends on whether Γ_A is larger or smaller than Γ_B .
5. How does the PM realize the linearly increasing potential term? What are the relation between α in Eq. (S11) and the modulation profile of the PM?

6. In the first paragraph on page 11, the author mentioned that since the upper band is amplified while the lower one is attenuated, what is the ratio of amplitude after amplified/attenuated during an adiabatic loop along the upper/lower band?

7. The wave amplitude governed by Eq. (S9) should decay exponentially in time, which is different from the numerical simulations and experiments presented in Fig. S2 (not exponentially decay). Is there any gain mechanism introduced in the simulations and by the EDFA?

Response to Reviewers and List of Changes Made

Reviewer 1:

1. Reviewer: In this paper, the authors demonstrate non-Abelian effects in non-Hermitian systems with dissipative couplings. They have implemented the dissipatively coupled resonators using time-multiplexed photonic optical cavities. Then they compared the experimental results with theoretical calculations based on the Lindblad equation and matrix-valued modified Wilson lines.

I find the paper very interesting, and I think it contains new physics and its clear demonstration. It is also very timely as non-Abelian properties in topological systems have recently drawn significant attention in the related fields. In addition, the proposal in the paper is original as it proposes dissipative coupling as a new way to demonstrate non-abelian nature, which has not been reported before as far as I am aware. The conclusions are well-supported by the experimental and theoretical results. Therefore, I recommend that the paper be accepted for publication in Nature Communications after addressing the following comments.

We would like to thank the reviewer for the careful review of our manuscript, the positive and constructive feedback, and his/her recommendation for publication.

2. Reviewer: Regarding the 2D honeycomb system Fig. 5, can the authors comment on how to implement it experimentally? Whereas the 1D SSH system can be more easily understood as the pulses are separated by the same interval in the time domain, extending the idea to 2D lattice systems may not be straightforward.

We thank the reviewer for this important remark. We have added a new section, titled “Possible Implementation of a Honeycomb Lattice in a Time-Multiplexed Resonator Network,” to the Supplementary Information. In this section, we show that it is possible to implement a honeycomb lattice with periodic boundary conditions using six delay lines. In this regard, such an implementation could be considered as an extension of our previous experiments on the 2D Harper-Hofstadter model in [Leefmans, et al. *Nat. Phys.* (2022)] where we used four delay lines to provide the necessary couplings in the lattice.

Fig. 1: Possible Implementation of a Honeycomb Lattice with a Time-Multiplexed Resonator Network. (a) A time-multiplexed network with six delay lines, which can be used to implement a dissipatively coupled honeycomb lattice with PBCs along both directions. (b) The mapping between the network in (a) and a honeycomb lattice. The coloration of the couplings corresponds to the coloration of the delay lines.

Here we summarize how to implement a hexagonal lattice in a time-multiplexed network. In Fig. 1, we present a time-multiplexed network capable of implementing a hexagonal lattice with periodic boundary conditions along both directions. In this network, the $\pm 1T_R$ delay lines implement couplings along the vertical direction, and the $\pm N_x T_R$ delay lines implement couplings along the horizontal direction. By selectively suppressing certain couplings along the vertical direction with the intensity modulators in the $\pm 1T_R$ delay lines, the couplings of the network may be mapped to those of a hexagonal lattice. Moreover, in this configuration, the $\pm N_x T_R$ automatically enable implementing a periodic boundary condition along the horizontal direction. Adding additional $\pm(N_x-1)T_R$ delay lines to the network enables implementing a periodic boundary condition along the vertical direction as well.

3. Reviewer: As shown in Fig. 5d, the non-Abelian nature can be realized with the 2D honeycomb lattice. However, it is unclear whether the non-commuting operation can also be realized in the dissipative 1D SSH lattice. Can the authors comment on whether the having a 2D system is necessary to realize well-defined non-commuting operators and whether it was chosen for a more effective presentation?

We would like to thank the reviewer for bringing up this important point. The underlying requirement for non-Abelian gauge fields in the reciprocal space is a Wilczek-Zee operator that does not commute at different points of the reciprocal space. However, to have non-Abelian “operators” one would need two separate paths that start and stop at the same point on the Brillouin zone. As the reviewer correctly alluded to, a 1D dissipative SSH can be used to observe signatures of the non-Abelian gauge fields (as we demonstrated experimentally via different Zak phases for upper and lower dissipative bands). However, to observe non-Abelian “operators”, one would need a lattice that is at least 2-dimensional. Our simulations in Fig. 5 show such non-Abelian operators in a dissipative honeycomb lattice.

We have made the following changes to the manuscript to be more clear about these important points (For the convenience of the reviewer, we are including these revised sections here):

- We have modified the last paragraph on page 3 in the Introduction: “Our simulations involving a two-dimensional honeycomb lattice illustrate how such dynamics lead to non-Abelian operators acting on the reciprocal space.”
 - We have added an explanation in this regard in the last paragraph of page 10 (before the conclusion): “So far, we showed that the geometric properties of dissipative bands in one-dimensional lattices involve the full matrix-valued Wilczek-Zee connections associated with the Bloch eigenstates in such systems. In this respect, our experimentally measured geometric phases provide signatures of non-Abelian effects in dissipative topological lattices. However, to define intersecting closed loops in the reciprocal space that are necessary for non-Abelian operators, one needs to consider lattices extending over more than one dimension.”
4. **Reviewer:** In the bottom panel of Fig. 4b, please explain what two vertical black arrows mean. As written in the caption, they seem to be related to the off-diagonal Wilczek-Zee connections, but it is not clear how the direction is determined and the physical meaning of the direction. Can the direction be associated with the signs of the off-diagonal terms?

The black dotted arrows indicate the exchange of population among the dissipative bands. As the lower band eigenstate $|\psi_{-}\rangle$ is initially excited at $k = 0$, the off-diagonal elements of the Wilczek-Zee operator results in a nonzero transfer of the population to the upper state (associated with $|\psi_{+}\rangle$). This first population transfer is shown in the inset as an upward black arrow. As the system undergoes Bloch oscillations (i.e. the Brillouin zone is swept from $k = 0$ to $k = 2\pi$), eventually part of the population in the upper band (which is stronger than the population in the lower band because of relative amplification associated with the upper band) returns to the lower band. This second transition, which overwhelms the original population in the lower band, is shown in the inset as a downward black arrow. The sign of the off-diagonal terms in the Wilczek-Zee matrix only affects the phase associated with the population exchanges between the bands. Since the direction of the black dotted arrows shown in Fig. 4 determine the absolute values of such population exchanges, they are expected to be independent of the sign of the off-diagonal elements in the Wilczek-Zee connection.

To clarify this, we have now added these descriptions to the caption in Fig. 4 of the manuscript:

“The left and right black dotted arrows represent the transfer of Bloch eigenstate populations from the lower band to the upper one and vice versa, respectively.”

5. Reviewer: A minor comment: please define the abbreviation of BO in the main text when it appears first for readers who are not familiar.

We thank the reviewer for pointing this out. We have now added this definition in the paper. Please see page 5, second paragraph, line 15 from the top: “This type of evolution results in periodic revivals in the dynamics of the optical fields across the lattice which are known as Bloch oscillations (BO).”

Reviewer 2:

1. Reviewer: In this manuscript, the authors study the presence of non-abelian effects in their previously presented resonator networks with dissipative couplings. Non-abelian effects are rare effects that could have an impact in topological quantum computing. This study is not the first study of non-abelian effects in topological photonics platforms, but it is the first in a purely dissipative topological photonic platform. The measurements are convincing, and the figures are excellent.

We would like to thank our referee for his/her careful review of our manuscript and the constructive comments. We earnestly believe that the clarity of our manuscript has greatly benefited because of his/her suggestions. We further thank our reviewer for finding our work novel and our measurements convincing. We provide a point-by-point response to his/her comments below.

2. Reviewer: However, I have two concerns:

The overlap with the authors’ earlier work. Both the experimental set up and the results show a very strong overlap with earlier works from the authors. For instance, the results presented up to Fig. 3 in the current measurement seem to have already been presented in Ref. 38, except for the

direct measurement of the Zak phase (however the measurement of the Zak phase just confirms that the results in Ref. 38 were indeed topological, a claim already made in Ref. 38).

It looks like the novelty lies in Fig. 4, where the authors show the modification of the geometric phase of the lower band associated with the excitation of the eigenstate of lower band of the dissipative SSH. If I understand correctly, these effects were always there, including in Ref. [38], however the new measurements of the geometric phase reveal the underlying non-abelian effects. Revealing the presence of non-abelian effects is a valid and important result, but at the moment it is just hard to understand whether this is just a deeper look into previously observed effects or whether the physical effects observed here are different. This needs to be clarified, as well as the explicit differences in the experimental setup and their importance.

We would like to summarize here the main results of our work, which clearly distinguishes the current manuscript from our previous work in Ref. [38]:

- Here, we measure *nontrivial geometric phases* associated with different windings and different bands of the dissipative SSH model. The 1D experiments in Ref. [38] were focused on the eigenvalue measurements and the associated band-structure together with topological phase transition from trivial to topological regimes. To measure the geometric phases, we have made two important modifications to the setup in Ref. [38]: 1) The present work uses a homodyne detection scheme to resolve the phase information encoded in the optical fields. 2) To introduce momentum space dynamics in the lattice, we use the Bloch oscillation (BO) technique. As discussed in Supplementary Part 2, this can be implemented by a discrete pulse to pulse phase difference in our system. We utilized a phase modulator in the main cavity to achieve this (please see Fig. 1).
- In this work, we find that the two upper and lower bands associated with the dissipative SSH model display different geometric phases. We show that this intriguing response can be explained by employing the generalized version of the Berry-Zak phases which involve the full matrix-valued Wilczek-Zee connection $\mathbf{A}_{p,q}$ defined in Eq. (2). Since the resulting modified Wilson line operator $\widehat{\mathcal{W}}'$ involves significant population exchange between the two bands involved in the system, one cannot reduce this description to a scalar Berry-Zak phase description.
- Finally, we generalize these notions to 2D dissipatively coupled lattices and study the dissipative honeycomb lattice as an example. Here, our numerical simulations that are

presented in Fig. 5 shows how the closed-loop dynamics in the reciprocal domain can involve non-Abelian operators.

To better reflect these distinctions with our earlier experiments, we have now made the following changes to the manuscript:

- Abstract: “Here, we show that Bloch eigenstates associated with lattices with dissipatively coupled elements exhibit geometric properties that cannot be described via scalar Berry phases, in sharp contrast to conservative Hamiltonians with non-degenerate energy levels. This unusual behavior can be attributed to the significant population exchanges among the corresponding dissipation bands of such lattices. Using a one-dimensional example, we show both theoretically and experimentally that such population exchanges can manifest themselves via matrix-valued operators in the corresponding Bloch dynamics. In two-dimensional lattices, such matrix-valued operators can form non-commuting pairs and lead to non-Abelian dynamics, as confirmed by our numerical simulations.”
- Page 5, second paragraph: “It should be emphasized that the setup used here has two important distinctions compared to the one used in [38] that enables us to experimentally investigate geometric properties and non-Abelian signatures associated with topological dissipative bands. First, the present work uses a homodyne detection scheme to resolve the phase information encoded in the optical fields, as necessary to measure geometric phases. Second, to probe the geometric properties of the dissipation bands, we apply a constant force in the reciprocal space which translates into a closed-loop evolution defined on the Brillouin zone in the Bloch-momentum space. This type of evolution results in periodic revivals in the dynamics of the optical fields across the lattice which are known as Bloch oscillations (BO). To achieve this, we implement the Hamiltonian $\hat{H}_{BO} = \mathbf{F} \cdot \hat{\mathbf{r}}$, where \mathbf{F} represents a constant effective force along the reciprocal lattice direction \mathbf{r} . In Supplementary Part 2 we show that this Hamiltonian can be approximated by a pulse-to-pulse linear phase gradient implemented by a phase modulator in the main cavity of our network.”
- Page 10 (before the conclusion): “So far, we showed that the geometric properties of dissipative bands in one-dimensional lattices involve the full matrix-valued Wilczek-Zee connections associated with the Bloch eigenstates in such systems. In this respect, our experimentally measured geometric phases provide signatures of non-Abelian effects in

dissipative topological lattices. However, to define intersecting closed loops in the reciprocal space that are necessary for non-Abelian operators, one needs to consider lattices extending over more than one dimension.”

3. **Reviewer:** Important concepts are missing from the paper and the writing is obscure. Most notably, the authors do not explain what non-abelian effects are or why are they important, despite the fact that this is the main point of their manuscript. On a related note, the abstract reads, “Here, we show that resonator networks with dissipative couplings can be governed by matrix-valued modified Wilson lines, leading to non-Abelian effects”, which is excessive jargon for the abstract. If this was a more specialized journal, maybe this would be OK, but for Nature Communications, I think this has to be fixed.

The authors launch into demonstrating Bloch oscillations without giving the reader any context on why they are doing that (also, the acronym BO is used without having been defined anywhere) .

The statements “... the shape of the diffraction pattern in this figure is qualitatively different from its conservative counter parts [61] because it involves constructive interference among the lattice elements due to the dissipative couplings involved” is unclear. Constructive interference is also a crucial aspect of discrete diffraction in conservative systems, so this statement on its own does not clarify anything.

We would like to thank the reviewer for suggesting these changes. We have now made the following changes per the reviewer’s suggestions:

- Abstract: “Here, we show that Bloch eigenstates associated with lattices with dissipatively coupled elements exhibit geometric properties that cannot be described via scalar Berry phases, in sharp contrast to conservative Hamiltonians with non-degenerate energy levels. This unusual behavior can be attributed to the significant population exchanges among the corresponding dissipation bands of such lattices. Using a one-dimensional example, we show both theoretically and experimentally that such population exchanges can manifest themselves via matrix-valued operators in the corresponding Bloch dynamics. In two-dimensional lattices, such matrix-valued operators can form non-commuting pairs and lead to non-Abelian dynamics, as confirmed by our numerical simulations.”
- Page 3, second paragraph: “On many occasions, these gauge fields belong to the Abelian class which give rise to closed-loop evolution operators in the parameter space that

commute with each other. This simple picture changes drastically in more complex scenarios which involve non-Abelian gauge fields where the corresponding operators along different paths are no longer commutative and can be utilized to obtain universal gates for topological quantum computing [46].”

- Page 5, second paragraph: “It should be emphasized that the setup used here has two important distinctions compared to the one used in [38] that enables us to experimentally investigate geometric properties and non-Abelian signatures associated with topological dissipative bands. First, the present work uses a homodyne detection scheme to resolve the phase information encoded in the optical fields, as necessary to measure geometric phases. Second, to probe the geometric properties of the dissipation bands, we apply a constant force in the reciprocal space which translates into a closed-loop evolution defined on the Brillouin zone in the Bloch-momentum space. This type of evolution results in periodic revivals in the dynamics of the optical fields across the lattice which are known as Bloch oscillations (BO). To achieve this, we implement the Hamiltonian $\hat{H}_{BO} = \mathbf{F} \cdot \hat{\mathbf{r}}$, where \mathbf{F} represents a constant effective force along the reciprocal lattice direction \mathbf{r} . In Supplementary Part 2 we show that this Hamiltonian can be approximated by a pulse-to-pulse linear phase gradient implemented by a phase modulator in the main cavity of our network.”

We further thank the reviewer for pointing out the discussion about discrete diffractions. In this statement, we meant to emphasize the distinction between dissipative diffraction patterns that we observe in our lattice versus the ordinary discrete diffractions that would occur in lattices with dispersive couplings. In the case of dispersive couplings, which are represented by imaginary coupling coefficients in the system Hamiltonian, the interferences among different lattice sites oscillate between constructive and destructive interference. This leads to the familiar nulls in the diffraction patterns observed in such systems. Mathematically, this is described by the zeros of the Bessel functions of the first kind $J_n(x)$ which form the analytical solutions for the wave amplitudes in these lattices. The situation changes drastically when the lattice involves dissipative couplings which are represented by real coupling coefficients in the evolution equations. As detailed in Part 7 of the Supplementary, the analytical solutions to these equations are instead given by the modified Bessel functions of the first kind $I_n(x)$ multiplied by exponentially decaying factors which only have nulls at the origin. In other words, here, the interferences among different sites

are always in-phase and therefore do not lead to intensity nulls. These theoretical predictions are further corroborated with our experimental measurements reported in Fig. 2 and Supplementary Fig. 4.

To clarify these aspects, we have now revised the statement in the main text and added the above discussion in Part 7 of the Supplementary: “We emphasize that the shape of the diffraction pattern in this figure is qualitatively different from its conservative counterparts [63] due to the dissipative couplings involved (see Supplementary Part 7).”

4. Reviewer: So, in summary, the paper would need to be written in a more clear language, avoiding ambiguities and being upfront about the novelty of the paper, as well as avoiding excessive jargon and giving the reader enough context.

We thank the reviewer for the constructive comments based on which we have made substantial changes to the manuscript. We hope that the reviewer will find the new version of our manuscript suitable for publication.

Reviewer 3:

1. Reviewer: The authors construct a one-dimensional pulse array with a splitter, a combiner, and properly chosen delay lines in this work. The delay lines introduce the jump operator, and the Lindblad master equation describes the dynamics of such a system. The authors first demonstrate the Bloch oscillation with a phase gradient introduced by a phase modulator (Fig. 2) and then measure the Zak phase inside a dissipative SSH model (Fig. 3). Here, the higher band is amplified while the lower band is attenuated. Due to the presence of the Wilczek-Zee connection matrix, there is some possibility that a mode can jump to the higher band even if it is initially excited at the lower band, and vice versa. Thus, if one measures the Zak phase of the lower band, one will unavoidably get the information of the higher band (Fig. 4). This is, technically speaking, not a non-abelian effect. Figure 5 tries to provide some information about how a non-abelian effect is possible. The title, abstract, and introduction try to argue that this system is non-abelian, but as said before, with no clear evidence to demonstrate the non-abelian behavior. Figure 5 finally returns to the focus (non-abelian) but with no experimental evidence, not even numerical simulations. Thus, I will not support its publication.

We thank the reviewer for the careful review of our manuscript. Here we provide detailed responses to the questions and concerns of the reviewer.

The underlying requirement for non-Abelian dynamics in the reciprocal space is a Wilczek-Zee operator that does not commute at different points of the reciprocal space [Wilczek, F. & Zee, PRL 52, 2111]. However, to define non-Abelian “operators” one would need two separate paths that start and stop at the same point on the Brillouin zone. As the reviewer correctly alluded to, a 1D dissipative SSH model can be used to observe only *signatures* of the non-Abelian gauge fields. In our experimental demonstration involving the dissipative SSH model, we find that the two upper and lower bands display different geometric phases. We show that this intriguing response can be explained by employing the generalized version of the Berry-Zak phases which involve the full matrix-valued Wilczek-Zee connection $\mathbf{A}_{p,q}$ defined in Eq. (2). Since the resulting modified Wilson line operator \widehat{W} describes significant population exchange between the two dissipation bands involved, one cannot reduce this behavior to a scalar Berry-Zak phase description. Such matrix-valued Wilson line operators are known as precursors for non-Abelian dynamics. In this respect, even though a 1D system like the dissipative SSH model implemented in our experiments does not allow for defining two different closed loops in its Brillouin zone for defining non-Abelian operators, it can nevertheless provide *signatures* of the underlying modified Wilson line operator and its *nonscalar* nature. However, to observe non-Abelian “operators”, one would inevitably need a lattice that is at least 2-dimensional. This is why in our manuscript we study the more general case of a 2D honeycomb lattice. We would like to emphasize here that the results presented in Fig. 5 are in fact *numerically simulated results* which clearly indicate the existence of non-Abelian operators in such a setting.

To clarify these points, we have made the following changes to the manuscript:

- Abstract: “Here, we show that Bloch eigenstates associated with lattices with dissipatively coupled elements exhibit geometric properties that cannot be described via scalar Berry phases, in sharp contrast to conservative Hamiltonians with non-degenerate energy levels. This unusual behavior can be attributed to the significant population exchanges among the corresponding dissipation bands of such lattices. Using a one-dimensional example, we show both theoretically and experimentally that such population exchanges can manifest themselves via matrix-valued operators in the corresponding Bloch dynamics. In two-

dimensional lattices, such matrix-valued operators can form non-commuting pairs and lead to non-Abelian dynamics, as confirmed by our numerical simulations.”

- Page 3, last paragraph: “To do so, we experimentally measure nontrivial geometric phases and demonstrate signatures of non-Abelian effects in a dissipatively-coupled network of time-multiplexed photonic resonators.” and last sentence “Our simulations involving a two-dimensional honeycomb lattice illustrate how such dynamics lead to non-Abelian operators acting on the reciprocal space.”
- Page 10 (before the conclusion): “So far, we showed that the geometric properties of dissipative bands in one-dimensional lattices involve the full matrix-valued Wilczek-Zee connections associated with the Bloch eigenstates in such systems. In this respect, our experimentally measured geometric phases provide signatures of non-Abelian effects in dissipative topological lattices. However, to define intersecting closed loops in the reciprocal space that are necessary for non-Abelian operators, one needs to consider lattices extending over more than one dimension.”

2. Reviewer: Figure 1. Need more details on why the combination of a splitter, a combiner, and the other two intensity modulators (IMs) corresponds to the jump operator. Supplementary Material Sec. 1 directly gives the results but has not explained why. For example, what is the relation between the modulation profile at the IMs and Γ ? How is the IM programmed and synchronized to realize the dimerized coupling of the SSH model?

In the new version of the Supplementary Information, we have added additional details in sections titled “Relationship Between the Master Equation and Dissipative Delay Line Couplings” and “Calibration Procedure”. These sections describe how the modulators in our setup are calibrated, explain the mathematical relationship between the modulation of the intensity modulators and the coupling strength Γ , and explain how we realize the dimerized couplings of the SSH model with our delay lines. Furthermore, we have added an additional section, “Dissipative and Conservative Coupling,” to the Supplementary Information, which summarizes the differences between dissipative and conservative couplings.

Here we summarize, from an experimental perspective, why the combination of the splitter, the combiner, and the intensity modulators enables us to implement the dynamics described by the

jump operators presented in Supplementary Information Part 1. We also comment on how our delay line intensity modulators can realize the staggered couplings of the SSH lattice.

Physically, the jump operators in the Lindblad master equation (Eq. 1) describe dissipative couplings between the sites of a lattice. These couplings occur when light from one site couples into a reservoir and then couples back from the reservoir to another site in the lattice. The splitter/modulator/coupler combination essentially acts as an intermediate reservoir that links two pulses in our time-multiplexed network and produces dissipative couplings between them. The splitter taps off a portion of the power in each pulse and passes it through the intensity modulator. This modulator controls the amount of that pulse that can pass through the delay line, and hence it controls the coupling strength of the coupling that occurs through the reservoir. Finally, the coupler combines the light from one pulse to another pulse in the setup, which accomplishes the dissipative coupling.

To implement the staggered couplings of the SSH model, we utilize the 10 GHz bandwidth of our intensity modulators to switch the coupling strengths of our delay lines from pulse to pulse. Each of the delay line intensity modulators is biased to minimum throughput. Then, as each pulse passes through the intensity modulator, we send an RF signal pulse to the IM that dictates the coupling strength of the associated optical pulse to its nearest neighbors. By modulating the intensity modulators in the delay line with alternating strong and weak RF pulses, we can implement the alternating strong and weak couplings of the SSH model (e.g., the coupling from pulse #1 to pulse #2 can be stronger than the coupling between pulse #2 and pulse #3, and so on).

3. Reviewer: The third paragraph on page 5. The abbreviation BO appears without definition. A few more words about what Bloch Oscillations are might be better here. The full name Bloch Oscillation appears at the end of the first paragraph on page 7.

We thank the reviewer for pointing this out. We have now added a description of what Bloch oscillations are and defined the abbreviation. Please see page 5 of the main text, last section of the second paragraph: “Second, to probe the geometric properties of the dissipation bands, we apply a constant force in the reciprocal space which translates into a closed-loop evolution defined on the Brillouin zone in the Bloch-momentum space. This type of evolution results in periodic revivals in the dynamics of the optical fields across the lattice which are known as Bloch oscillations (BO).”

4. Reviewer: The second paragraph on page 7. “In this sense, the multiband open ...”. Why multiband? Or just only two bands?

We have now changed “multiband” to “two-band” in the main text as per our reviewer’s suggestion.

5. Reviewer: Figure 3(b) shows that $\Gamma_A = \Gamma_B$ corresponds to a trivial case where there is no gap between the two bands in Fig. 3a. Figures 3(c) and (d) correspond to topological cases where there is a gap (for imaginary energy). Why here is $\pm \pi/2$, not π and 0 for the typical Hermitian SSH model. Meanwhile, in a standard Hermitian SSH model, whether a Hamiltonian is topological or trivial depends on whether Γ_A is larger or smaller than Γ_B .

There are two questions here: “Why here is $\pm \pi/2$, not π and 0 for the typical Hermitian SSH model?” This is because for the SSH model (both the conservative Hamiltonian as well as the dissipative SSH implemented here) it is the difference between the Zak phases of the two dimerizations D_1 and D_2 depicted in Fig. 3 that is a topological invariant. In fact, the absolute Zak phases associated with each of these dimerizations depend on the choice of a unit cell in the lattice. Below we mathematically show this property.

Assume that $|\psi(x)\rangle = e^{ikx}|u_k(x)\rangle$ is the Bloch eigenstate associated with the SSH Hamiltonian (similar results apply to the effective Hamiltonian in the case of the dissipative SSH model). In this case, the Zak phase associated with this Bloch eigenstate is equal to

$$\phi_Z = \int_{-\pi/a}^{\pi/a} i \left\langle u_k \left| \frac{\partial}{\partial k} \right| u_k \right\rangle dk.$$

Here, $\pm \pi/a$ determine the boundaries of the Brillouin zone in the reciprocal space. Under a shift in the coordinates $x' = x + d$, this Bloch eigenstate will transform to $e^{ikx'} e^{-ikd} |u_k(x' - d)\rangle$. Therefore, the new Zak phase will transform according to

$$\phi'_Z = \int_{-\pi/a}^{\pi/a} i \left\langle e^{ikd} u_k \left| \frac{\partial}{\partial k} \right| e^{-ikd} u_k \right\rangle dk = \phi_Z + \frac{2\pi}{a} \times d.$$

Therefore, as evident from the second equation, the Zak phase is now shifted by $2\pi/a \times d$. For instance, by choosing $d = -a/4$ one can switch between $\phi_Z = 0, \pi$ and $\phi'_Z = -\pi/2, \pi/2$.

However, since the Zak phases associated with the two different dimerizations D_1 and D_2 shift in an identical manner, their difference $\phi_{Z,D_1} - \phi_{Z,D_2}$ remains invariant and is always equal to $\pm\pi$ for a topologically nontrivial SSH model.

The second question by the reviewer states “Meanwhile, in a standard Hermitian SSH model, whether a Hamiltonian is topological or trivial depends on whether Γ_A is larger or smaller than Γ_B .”

To answer this, we should note that the definition of being topological or trivial that the reviewer is alluding to concerns the existence/lack of edge states when an SSH lattice is terminated. Here, however, we are characterizing the bulk properties of the SSH lattice, which is independent of how the lattice is terminated. To do so, we measure the differential Zak phase which is defined as $\phi_Z = \phi_{Z,D_1} - \phi_{Z,D_2}$. As explained in the discussion above, unlike the Zak phases associated with the two dimerizations of the SSH model, i.e. ϕ_{Z,D_1} and ϕ_{Z,D_2} which are not invariant, the differential Zak phase is in fact a topological invariant that describes the bulk topology of the lattice. In this sense, the SSH lattice implemented in our experiments is topologically trivial when $\Gamma_A = \Gamma_B$ and nontrivial when $\Gamma_A \neq \Gamma_B$, corresponding to $\phi_Z = 0$ and $\phi_Z = \pi$, respectively.

To clarify these aspects, we have now added the discussions mentioned above to the Supplementary. Please see Supplementary Part 8, last paragraph.

6. Reviewer: How does the PM realize the linearly increasing potential term? What are the relation between α in Eq. (S11) and the modulation profile of the PM?

We thank the reviewer for their question. We discussed how the phase modulator approximates the linearly increasing potential term in Part 2 of the Supplementary Information. We have modified this section to improve its clarity, and we have summarized our updated analysis here.

First, we consider the effective Hamiltonian that describes dissipatively coupled Bloch oscillations. We can write such a Hamiltonian as

$$H = H_C + H_{BO} = \sum_n [-iJ|n+1\rangle\langle n| - iJ|n\rangle\langle n+1| + i\gamma|n\rangle\langle n|] + \sum_n \alpha n|n\rangle\langle n|,$$

and the equation of motion for such a system can be written as

$$\partial_t |\psi\rangle = i(H_C + H_{BO})|\psi\rangle.$$

We can formally integrate this expression and then apply the Baker-Campbell-Hausdorff formula to separate the dynamics due to the couplings and the dynamics due to the linear potential. This approximation introduces error on the order of $(\Delta t)^2$ and results in an equation of the form

$$|\psi(t + \Delta t)\rangle = e^{iH_C\Delta t} e^{iH_{BO}\Delta t} |\psi(t)\rangle.$$

We then make a second approximation by expanding the first exponential around $\Delta t = 0$. This again introduces error on the order of $(\Delta t)^2$, and it results in the expression

$$|\psi(t + \Delta t)\rangle = (1 + iH_C\Delta t) e^{iH_{BO}\Delta t} |\psi(t)\rangle.$$

This equation is what we implement with our experimental setup. To see why, let us first focus on the exponential term. This term is explicitly given by

$$e^{iH_{BO}\Delta t} = e^{i\sum_n \Delta t \alpha n |n\rangle\langle n|}.$$

We operate with this term on the state vector $|\psi(t)\rangle$, which we decompose into the position basis as

$$|\psi(t)\rangle = \sum_m c_m(t) |m\rangle.$$

The result is

$$e^{i\sum_n \Delta t \alpha n |n\rangle\langle n|} |\psi(t)\rangle = e^{i\sum_n \Delta t \alpha n |n\rangle\langle n|} \sum_m c_m(t) |m\rangle = \sum_m e^{i\Delta t \alpha m} c_m(t) |m\rangle.$$

Therefore, we see that to implement this exponential operator, we just need to multiply the sites (pulses) of our lattice by a linear phase ramp. We do this with the phase modulator in the main cavity. Note that this equation also shows how the phase applied to each site in our network is related to the parameter α from Eq. S11. In other words, the pulse-to-pulse phase difference necessary for this realization is given by $\Delta\varphi = \alpha\Delta t$.

Next, let us consider the terms in parentheses, $(1 + iH_C\Delta t)$. We can expand these terms as follows:

$$1 + H_C\Delta t = (1 - \gamma\Delta t) |n\rangle\langle n| + J\Delta t |n+1\rangle\langle n| + J\Delta t |n\rangle\langle n+1|.$$

The first term is implemented by the cavity feedback and the roundtrip losses. The second two terms are implemented by the delay line couplings.

Therefore, we see that, combined with the cavity feedback and the delay line couplings, applying a linear phase sweep to the phase modulator in the main cavity, enables us to implement dissipative Bloch oscillations in our system with error on the order of $(\Delta t)^2$. The timescale Δt is dictated by the roundtrip time of our cavity, which is ~ 256 ns, and the validity of this approximation for the

present experiments is justified by the correspondence between our measurements and the theoretical predictions of the full Hamiltonian without the approximations discussed above.

7. Reviewer: In the first paragraph on page 11, the author mentioned that since the upper band is amplified while the lower one is attenuated, what is the ratio of amplitude after amplified/attenuated during an adiabatic loop along the upper/lower band?

To answer this question, one must insert the initial Bloch eigenstate $|\psi(x, 0)\rangle = e^{ik_0x}|u_{k_0}(x)\rangle$ into the equation of motion given in Eq. (18) in the Supplementary with the Hamiltonian $\hat{H}_t = \hat{H}_{eff,0} + \hat{H}_1$, where $\hat{H}_{eff,0}$ denotes the effective SSH Hamiltonian in the absence of the effective force and $\hat{H}_1 = \sum_n \alpha n|n\rangle\langle n|$ is the uniform effective force applied on the lattice (responsible for Bloch oscillations). By assuming a generic solution $|\psi(x, t)\rangle = e^{-\gamma(t)}e^{i\phi(t)}e^{i(k_0-vt)x}|u_{k_0-vt}(x)\rangle$, one can find the total attenuation experienced by the initial Bloch eigenstate after one full cycle of traversing along the upper/lower band in the Brillouin zone

$$\gamma_{\pm} = \frac{a}{\alpha} \int_{k_0}^{k_0-2\pi/a} E_{\pm}(k) dk,$$

where $E_{\pm}(k)$ represent the Bloch eigenvalues associated with upper/lower bands. In our dissipative SSH system we have $E_{-}(k) = -E_{+}(k) = -\sqrt{\Gamma_A^2 + \Gamma_B^2 + 2\Gamma_A\Gamma_B \cos k}$. Therefore, the ratio between the amplitudes associated with the upper/lower Bloch eigenstates at the end of a Bloch period is given by $e^{2\frac{a}{\alpha} \int_{k_0}^{k_0-2\pi/a} E_{+}(k) dk}$.

We have now added these results to the Supplementary. Please see Supplementary Part 8, second last paragraph.

8. Reviewer: The wave amplitude governed by Eq. (S9) should decay exponentially in time, which is different from the numerical simulations and experiments presented in Fig. S2 (not exponentially decay). Is there any gain mechanism introduced in the simulations and by the EDFA?

We appreciate the reviewer's comment. In all cases regarding Bloch oscillation measurements and simulation results, we normalize the power in every round trip of the network to provide a more distinct visualization of the field intensities.

We have now clarified this representation in the figure captions associated with Fig. 2 of the main text and Fig. S4 of the Supplementary.

REVIEWER COMMENTS

Reviewer #1 (Remarks to the Author):

I am satisfied with the responses to my comments and changes in the manuscript. As I mentioned earlier, I believe the paper contains an intriguing proposal for photonic topological system with non-Abelian effects and exciting simulation and experimental results. The paper has also been improved in terms of presentation and clarity. Therefore, I recommend that it be accepted for publication in Nature Communications.

Reviewer #2 (Remarks to the Author):

The authors have added several paragraphs to clarify the nature of non-abelian effects, the importance of showing Bloch oscillations, and the differences with their previous work. While I still find that there is a bit too much jargon, I can recommend publication.

An interesting point is that raised by Reviewer 3, regarding the lack of direct experimental proof of non-abelian effects. In my opinion, while Reviewer 3 is correct in highlighting the lack of direct proof, the argument presented by the authors in their response is a valid one: they experimentally show signatures of potential non-abelian effects in the 1D case, and numerical simulations indicating non-abelian effects in the 2D systems. The combination of these two evidences should be enough to justify publication bearing in mind the novelty of the paper.

Reviewer #3 (Remarks to the Author):

The authors have addressed my concerns about technical issues in their report. Meanwhile, the authors have also weakened their claims in the abstract and introduction, which now match their contributions. I understand that the experimental demonstration of Fig. 5 might be a challenge, however, the simulations in Fig. 5 should at least be improved to support the title. As has also been mentioned by the first referee, there should be an implementation of the system such as the honeycomb lattice (Fig. 1 in the reply), and full-wave simulations based on it should be performed (similar to Fig. 2 or Fig. S4). Meanwhile, the evolution of states on the Bloch sphere along different paths should also be provided. A single arrow is not enough and misleading (like a schematic drawing).

Response to Reviewers and List of Changes Made

Reviewer 1:

Reviewer: I am satisfied with the responses to my comments and changes in the manuscript. As I mentioned earlier, I believe the paper contains an intriguing proposal for photonic topological system with non-Abelian effects and exciting simulation and experimental results. The paper has also been improved in terms of presentation and clarity. Therefore, I recommend that it be accepted for publication in Nature Communications..

We would like to thank the reviewer for the careful review of our manuscript, the positive and constructive feedback, and his/her recommendation for publication.

Reviewer 2:

Reviewer: The authors have added several paragraphs to clarify the nature of non-abelian effects, the importance of showing Bloch oscillations, and the differences with their previous work. While I still find that there is a bit too much jargon, I can recommend publication.

An interesting point is that raised by Reviewer 3, regarding the lack of direct experimental proof of non-abelian effects. In my opinion, while Reviewer 3 is correct in highlighting the lack of direct proof, the argument presented by the authors in their response is a valid one: they experimentally show signatures of potential non-abelian effects in the 1D case, and numerical simulations indicating non-abelian effects in the 2D systems. The combination of these two evidences should be enough to justify publication bearing in mind the novelty of the paper.

We would like to thank the reviewer for the careful review of our manuscript, the positive and constructive feedback, and his/her recommendation for publication.

Reviewer 3:

Reviewer: The authors have addressed my concerns about technical issues in their report. Meanwhile, the authors have also weakened their claims in the abstract and introduction, which now match their contributions. I understand that the experimental demonstration of Fig. 5 might be a challenge, however, the simulations in Fig. 5 should at least be improved to support the title. As has also been mentioned by the first referee, there should be an implementation of the system such as the honeycomb lattice (Fig. 1 in the reply), and full-wave simulations based on it should be performed (similar to Fig. 2 or Fig. S4). Meanwhile, the evolution of states on the Bloch sphere along different paths should also be provided. A single arrow is not enough and misleading (like a schematic drawing).

We thank the reviewer for the careful review of our manuscript and the positive and constructive feedback.

As per the reviewer's suggestion, we have now provided a more comprehensive version of the simulations presented in Fig. 5 of the manuscript in Fig. S10 of the Supplementary. In the new figure, we have shown the evolution of the states on the Bloch sphere with more details. For the convenience of the reviewer, we are also including this new figure here.

Figure 1: Non-Abelian dynamics in a dissipative honeycomb lattice. Simulation results showing the evolution of states on the Bloch sphere in a dissipative honeycomb lattice. It is evident that different operators corresponding to momentum-space dynamics along different closed loops C_1 and C_2 within the Brillouin zone do not commute and display non-Abelian behavior.

In addition, we have included a new Fig. S11 in the Supplementary as per reviewer's suggestion. This figure, similar to Fig. 2 and Fig. S4, visualizes the discrete evolutions of the state in the dissipative honeycomb lattice under different momentum-space operations defined by the closed loops C_1 and C_2 in the Brillouin zone. For the convenience of the reviewer, we are also including this new figure here.

Figure 2: Non-Abelian state evolutions in a dissipative honeycomb lattice. Simulation results regarding discrete state evolutions under different momentum-space operators corresponding to different closed-loops C_1 and C_2 within the Brillouin zone in a dissipative honeycomb lattice. The plots show amplitudes and phases of the fields associated with the A and B sites within a unit cell of the lattice (shown on the left panel).

We have also added a new sentence (highlighted in yellow) on page 13 of the manuscript to refer to these new results in the Supplementary.

REVIEWERS' COMMENTS

Reviewer #3 (Remarks to the Author):

The authors have addressed my questions tor comments with my satisfaction, and now I would like to recommend the manuscript for publication.